# HDDI: A Historical Data-Based Diffusion Imputation Method for High-Accuracy Recovery in Multivariate Time Series with High Missing Rate and Long-Term Gap

## Abstract

Multivariate time series data often face the challenge of missing values, which can impact the performance of subsequent tasks. Although some deep learning-based imputation methods perform well, they still struggle with insufficient training data due to high missing rate and long-term missing data. To address these challenges, we propose a Historical Data-based Multivariate Time Series Diffusion Imputation (HDDI) method. Unlike existing deep learning-based imputation methods, we design a historical data supplement module to match and fuse historical data to supplement the training data. Additionally, we propose a diffusion imputation module that utilizes the supplement training data to achieve high-accuracy imputation even under high missing rate and long-term missing scenario. We conduct extensive experiments on five public multivariate time series datasets, the results show that our HDDI outperforms baseline methods across five datasets. Particularly, when the data missing rate is 90%, HDDI improves accuracy by 25.15% compared to the best baseline method in the random missing scenario, and by 13.64% in the long-term missing scenario. The code is available at https://github.com/liuyu3880/HDDI_project.

## 1 Introduction

### 1.1 Background

Multivariate time series data is ubiquitous across various application domains, including transportation Liu et al. (2023); Wu et al. (2019), industry Chang et al. (2024), and healthcare Li et al. (2023b). However, this data often suffers from losses due to sensor failures, communication breakdowns, or unexpected malfunctions. Such missing data damages the interpretability of data Silva et al. (2012); Chauhan et al. (2022) and severely affects the performance of subsequent tasks, such as climate change research and patient monitoring. Therefore, accurate imputation of missing data is crucial.

To address the problem of missing data imputation, some studies propose algorithms based on statistics and classic machine learning Kreindler & Lumsden (2016); Little & Rubin (2019); Basharat & Shah (2009); Wang et al. (2006); Yu et al. (2016); Li et al. (2023a); Pujianto et al. (2019); Nelwamondo et al. (2007); Shu & Ye (2023). However, these methods rely on specific data assumptions Liu et al. (2023). For instance, linear interpolation methods typically assume a linear relationship between data points Kreindler & Lumsden (2016), while mean/median methods assume that the data follows a uniform distribution Little & Rubin (2019). The K-Nearest Neighbors (KNN) methods assume that data points have similar characteristics within their local neighborhoods Pujianto et al. (2019). When these assumptions do not hold, the accuracy of data interpolation can be significantly affected.

In contrast to statistics and classic machine learning-based algorithms, some recent studies propose deep learning-based imputation methods Cao et al. (2018); Yoon et al. (2017); Che et al. (2018); Cini et al. (2021); Suo et al. (2020). These methods leverage neural network models to extract data features and perform the imputation of missing values without relying on specific data assumptions. For example, some works use Recurrent Neural Networks (RNNs) Cao et al. (2018); Yoon et al. (2017); Che et al. (2018) to capture temporal dependencies in the data and use these dependencies for imputing missing values. Other methods Suo et al. (2020) capture dependencies between sequence elements through self-attention mechanisms. However, these methods are prone to encountering the issue of error accumulation (i.e., inference of missing values from inaccurate historical imputationLiu et al. (2019)).

Additionally, some research introduces imputation methods based on generative models. For example, methods Gong et al.; Miao et al. (2021); Luo et al. (2018); Yoon et al. (2018a) based on Generative Adversarial Networks (GANs) use the generator to estimate missing values and the discriminator to assess whether these estimates align with real data. Variational Autoencoders (VAEs) Fortuin et al. (2020) map data to the latent space through an encoder and then generate imputation results from this latent space using a decoder. Through adversarial training, the generator progressively improves the accuracy of missing data imputation, but they still suffer from training instability.

With the development of generative models, the study Sohl-Dickstein et al. (2015) proposes the diffusion model, which is more robust to different types of noise and missing data compared to traditional generative models. The diffusion model is applied to various applications, such as image processing Song et al. (2020), and audio signal processing Kong

et al.. Morever, as the diffusion models can avoid the error accumulation issues in RNN-based imputation methods and offer a more stable training process than generative adversarial networks, by using flexible architectures neural network architecture, the diffusion model is also applied to the imputation problem to achieve higher performance. Such as, CSDI Tashiro et al. (2021) is proposed to solve the data imputation problem based on the diffusion model. The diffusion model starts with randomly sampled Gaussian noise and progressively removes noise to transform the noisy data into imputed values.

## 1.2 CHALLENGES

Although a limited number of works begin to use diffusion models for data imputation and achieve promising results, their effectiveness remains limited in scenarios with high missing rate and long-term missing data:

**High missing rate data.** Since these deep learning models rely on self-supervised learning, they need to select a subset of the collected sparse measurement data as supervised data for training. This approach reduces the amount of extractable information as the training data decreases. In scenarios with a high missing data rate, methods based on diffusion models struggle to extract sufficient information, which affects the accuracy of imputing missing data.

**Long-term missing data.** Due to equipment failures or communication issues, data from certain devices may be missing for extended periods, a situation known as long-term missing dataPark et al. (2023). On one hand, long-term missing data leads to a high missing rate. On the other hand, the absence of data from faulty equipment may cause imputation results to overfit data from non-faulty equipment while underfitting data from faulty equipment, results in lower imputation accuracy.

## 1.3 CONTRIBUTIONS

To address the above challenges, we propose a novel imputation model: Historical Data-Based Multivariate Time Series Diffusion Imputation (HDDI). Unlike existing methods based on diffusion models, HDDI fully utilizes historical features by searching for similar data from historical records and combining it with current target observational data to create new training data. By incorporating historical data, HDDI provides sufficient training data for diffusion-based imputation algorithms, significantly improving the imputation accuracy of multivariate time series data in scenarios with high missing rate and long-term missing scenario. Specifically, our contributions are as follows:

**1) We propose HDDI based on a diffusion model:** By matching historical data to supplement the target observational data, HDDI addresses the challenges posed by high missing data rate and long-term missing scenarios. This approach resolves the issue where insufficient data features hinder the accurate prediction of missing data, providing more valuable information for model training and enhancing the accuracy of predicting missing data at the current time.

**2) We design a sliding window-based historical data matching and combination scheme:** By selecting multiple segments of historical data that best match the current time segment to supplement training data, we tackle the problem of insufficient training data in self-supervised learning. Additionally, we design a historical data fusion scheme, carefully considering the similarity and temporal correlations between target observational data and historical data. We utilize multiple historical data segments and fuse them with target observational data through normalization to ensure that the distribution of the fused training data approximates the range of true values.

**3) We design a diffusion-based imputation algorithm:** The diffusion model primarily consists of two processes: the diffusion process and the denoising process. The diffusion process trains a noise estimation model by simultaneously inputting the fused training data and some observational data, which provides rich feature information for the model's training. The denoising process gradually removes the noise estimated by the noise estimation model from random noise data, ultimately resulting in accurate imputed data.

**4) HDDI demonstrates excellent performance in experiments:** We evaluate the proposed HDDI with five state-of-the-art baseline methods under five multivariate time series datasets. The experimental result indicates that our proposed HDDI achieves the best performance compared to other baseline methods. Even in cases where 90% of the data was missing, our HDDI outperformed the best baseline method by 25.15%.

## 2 RELATED WORK

In this section, we introduce some existing deep learning-based time series imputation methods.

**1) Temporal Feature Extraction-Based Methods:** When handling time series data with Recurrent Neural Networks (RNNs), the study Che et al. (2018) proposes GRU-D, which estimates time series using a deep learning model and introduces the concept of time lag by employing hidden state decay to capture past features. M-RNN Yoon et al. (2018b) and BRITS Cao et al. (2018) both use bidirectional RNNs to incorporate bidirectional information flow, allowing them to learn patterns from the context around missing data and perform imputation. However, RNNs are prone to compounded errors Liu et al. (2019) during training, which accumulates with increasing sequence length, leading to reduced imputation accuracy.

To address these limitations, some research shifts to methods based on self-attention mechanisms, such as SAITS Du et al. (2023). These methods use self-attention mechanisms to directly model dependencies between sequence elements, reducing the impact of compounded errors and improving the model's ability to handle long sequences. However, such a method requires a large amount of high-quality training data. When the training data is insufficient, this model cannot achieve satisfactory imputation results.

**2) Generative-Based Methods:** Imputation methods based on generative models are gaining increasing popularity. For example, GP-VAE Fortuin et al. (2020) utilizes a deep variational autoencoder (VAE) to obtain latent representations and employs Gaussian processes in latent space to capture the global dynamics and structure of time series, providing an advantage in handling continuous missing data. Even in long-term missing scenarios, GP-VAE leverages its understanding of global dynamics to generate reasonable imputations. Similarly, SS-GAN Miao et al. (2021) employs semi-supervised learning mechanisms to guide the learning process of generators and discriminators. Nevertheless, these methods often face issues with interpolation accuracy caused by unstable training.

Thanks to the introduction of diffusion model Ho et al. (2020); Song et al. (2020), the study proposes CSDI Tashiro et al. (2021) which learns data distribution through conditional score-based diffusion model, gradually transforming noise into reasonable time series via denoising processes. CSDI improves imputation accuracy by incorporating observational information into the diffusion model using conditional data, achieving better performance compared to other existing time series imputation methods. However, practical scenarios often provide limited observational data, and a low sampling rate may hinder the effectiveness of missing data imputation.

Different from the aforementioned methods, our imputation model HDDI supplements training data by identifying and matching historical data with current target imputation data to address issues related to low sampling rate. Building on this, we propose a novel diffusion model that fuses target imputation data with matched historical data to achieve high-precision imputation of missing data.

# 3 PROBLEM FORMULATION

Multivariate time series data is represented as a three-dimensional tensor $\mathcal{X} \in \mathbb{R}^{I \times J \times K}$, where $I$ denotes the number of features, $J$ represents the number of time points, and $K$ indicates the number of data samples. For example, in air quality data, $I$ denotes the number of metrics, e.g., NO2, PM2.5, and CO, and $K$ denotes the number of sensing devices. In medical data, $I$ denotes the number of physiological indicators of ICU patients, e.g., Lactate, PaO2, and Glucose, and $K$ denotes the number of patients. However, due to cost reduction, equipment failures, or human errors, the recorded data tensor is highly sparse, as shown in Fig.1(a).

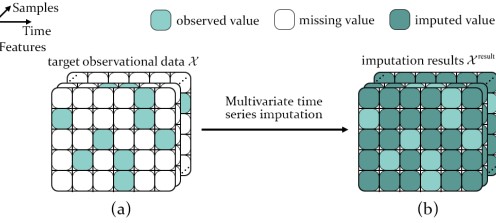

Figure 1: Sparse multivariate time series data tensor $\mathcal{X}$ and its imputation results $X^{\text{result}}$.

To indicate the positions of observed data, we define a mask tensor $\mathcal{M}$ as follows:

$$m_{i,j,k} = \begin{cases} 0, & \text{if } x_{i,j,k} \text{ is missing;} \\ 1, & \text{if } x_{i,j,k} \text{ is observed,} \end{cases} \tag{1}$$

where $m_{i,j,k} = 0$ indicates that the data point $x_{i,j,k}$ is missing, while $m_{i,j,k} = 1$ signifies that the data point $x_{i,j,k}$ exists, with $0 \leq i < I$, $0 \leq j < J$, and $0 \leq k < K$.

In this paper, we primarily focus on the multivariate time series imputation problem, aiming to accurately impute missing data in the target observational data $\mathcal{X}$ using an imputation model. This process involves treating the missing data as the imputation target to obtain complete imputation results $X^{\text{result}}$, as illustrated in Fig.1(b).

Since the true missing values are unknown, we lack supervised data to guide the training of the imputation model. Therefore, we adopt a self-supervised learning approach by dividing the target observational data into training data and supervised data. Through supervised data providing true values, the model is guided in learning how to estimate missing values during the training process. Specifically, we randomly sample a portion of the target observational data $\mathcal{X}$ to serve as supervised data $\mathcal{X}_{\text{sup}}$, while the remaining data is used as training data $\mathcal{X}_{\text{train}}$. Thus, $\mathcal{X} = \mathcal{X}_{\text{sup}} + \mathcal{X}_{\text{train}}$.

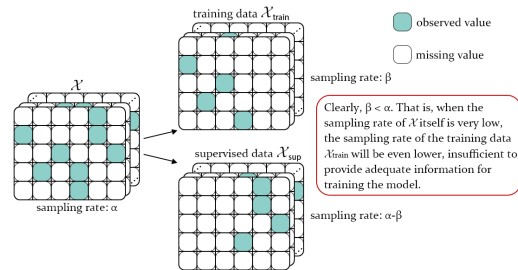

Figure 2: Self-supervised learning methods under low sampling rate.

For multivariate time series data, there are typically two scenarios of data missing:

**1) Random Missing (RM) Scenario:** Due to occasional communication failures (e.g., UDP packet loss) during data transmission, any sensor may experience data loss for one or multiple consecutive time points. In this scenario, the distribution of the missing data is random and unstructured.

**2) Long-term Missing (LM) Scenario:** In practical applications, sensor failures due to component malfunctions, battery interruptions, or exposure to harsh weather conditions and dust are common. As a result, there can be long-term missing data. As shown in Fig.3, all devices are operational before time $t_1$, allowing the initial data matrix $X$ to collect monitoring data from all devices. However, at time $t_1$, devices 2 and 4 fail, causing their data to become uncollectible thereafter, leading to missing entries in the corresponding row of matrix $X$ after time $t_1$.

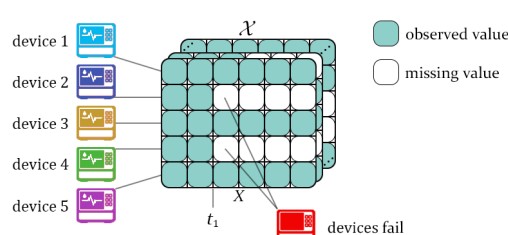

Figure 3: The Long-term Missing (LM) scenario leads to extended periods of missing data for certain attributes within the data tensor.

In summary, the training imputation model in the two main missing data scenarios mentioned above typically presents the following issues:

1) Both data-missing scenarios often involve excessively high missing data rate, leading to a severe shortage of training data. Specifically, the self-supervised learning approach reduces the amount of observational data used for model training, which negatively impacts the accuracy of data imputation.

2) In the LM scenario, during the training process of the imputation model, the model can only learn data distribution characteristics from the non-faulty devices and cannot capture data features from the faulty devices. This results in the imputation model overfitting to the data distribution of the non-faulty devices while underfitting the data feature of the faulty devices, leading to significant long-term missing data issues. Consequently, the imputation accuracy for data from faulty devices suffers.

To address these issues during the training of the imputation model, we seek supplementary training data from historical records.

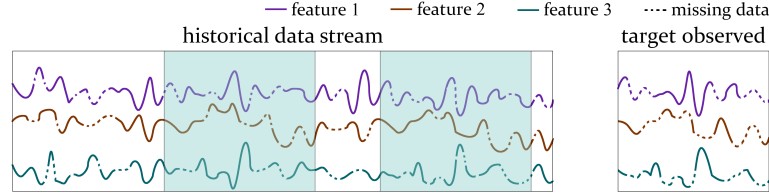

Fig.4 shows a visualization of a multivariate time series sample (e.g., sensor data collected from a device). The three differently colored curves represent the values of three features over time, with the dashed portions indicating missing values for those features at specific times. We observe that a sequence segment with missing data in the current time period often finds one or more similar historical segments (highlighted in blue boxes). Additionally, we note that the positions with missing data in the current time segment may not necessarily be missing in the historical data, allowing historical data to provide supplementary features for the current target observational data.

Figure 4: Illustration of the correlation of time series data over history.

Inspired by this observation, our HDDI aims to identify information from historical data that is similar to the target observational data and combine them with the training data $X_{\text{train}}$ to create a more comprehensive training data $X^{\text{fu}}$ for training subsequent diffusion imputation model. By fully utilizing historical data, the method provides more data features for the diffusion model, addressing issues of insufficient training data and long-term missing data, thereby enhancing imputation accuracy.

In the following sections, to facilitate the description of our proposed HDDI, we use a sample (e.g., the sensor data collected by a device) as an example, which can be represented as $X \in \mathbb{R}^{I \times J}$. Symbolic expressions are shown in Table 1 of Appendix 7.1 for clarity.

## 4 PROPOSED METHOD

Our proposed HDDI consists mainly of the Historical Data Supplement Module and the Diffusion Imputation Module as shown in Fig.5.

**Historical Data Supplement Module:** To address the issue of insufficient training data in high missing rate scenarios, we design a Historical Data supplement module. This module matches data from the historical data stream that is most similar to the target observational data and performs fusion to generate

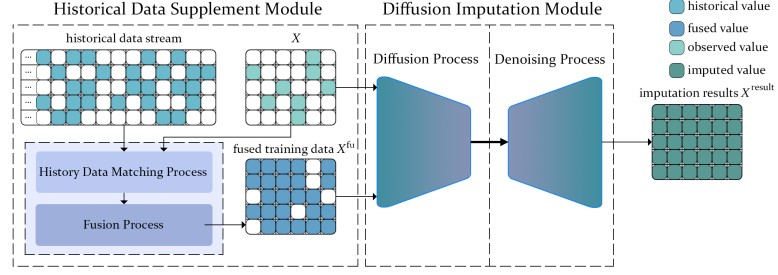

Figure 5: Graphical overview of the HDDI Model, demonstrating the two main modules: the Historical Data Supplement Module and the Diffusion Imputation Module.

fused training data $X^{\mathrm{fu}}$. This fused data can provide more comprehensive training data for training imputation model for missing data.

**Diffusion Imputation Module:** The diffusion imputation module takes both the target observational data $X$ and the fused data $X^{\mathrm{fu}}$ as inputs. It consists of two main components: the diffusion process and the denoising process. The diffusion process begins with the supervised data $X_{\mathrm{sup}}$ and progressively adds noise until the data becomes pure noise. This process trains a noise estimation model. The denoising process starts with random noise data and applies the noise estimation model to estimate and gradually remove the noise, generating new imputed data. This imputed data is then combined with the target observational data to produce the final imputation results $X^{\mathrm{result}}$. Next, we detail the design of each module.

## 4.1 HISTORICAL DATA SUPPLEMENT MODULE

Due to the high missing rate often encountered in multivariate time series data, which results in insufficient training data from random missing and long-term missing data problems, it is necessary to search historical data for segments that match the current target observational data $X$. This approach aims to provide more sufficient training data for subsequent imputation module. Therefore, we design the historical data supplement module to extract sufficient and useful information from historical data, ensuring adequate and accurate training data for the imputation module.

However, extracting sufficient and effective information from high missing rate data remains a challenging task. In practical applications, historical data also experiences a high missing rate. Consequently, merely extracting a segment of data from historical records to supplement the target observational data does not completely resolve the issues of insufficient training data and long-term missing data.

Therefore, we design a *Historical Data Matching Process* to match $K$ segments of data from historical records and a *Fusion Process* to fuse the selected segments of data with the train data $X_{\mathrm{train}}$ divided from $X$. Among these, the $K$ is a hyperparameter.

**Historical Data Matching Process:** To obtain sufficient supplement data, we need to identify $K$ historical data segments that best match the target observation data among the entire historical data stream. Therefore, we designed the *Historical Data Matching Process* based on the sliding window strategy.

As illustrated in Fig.6(a), given a current target observational data, denoted as $X \in \mathbb{R}^{I \times J}$, we slide a matching window across the historical data stream to find the most similar historical data segments. The sliding window starts from the first time slot and slides with the step size of 1. Therefore, for a historical data stream with $T$ time slots, the current target observational data will match with data from $T - J + 1$ historical time segments.

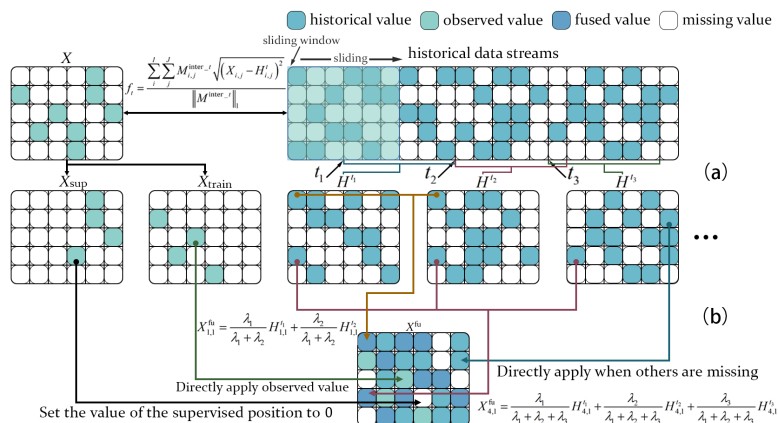

Figure 6: Matching historical data based on sparse historical data matrices.

During this process, we need to compute the similarity $s_t$ between the target observational data $X$ and each historical data segment $H^t$ selected by the sliding window, where $t = \{1, 2, ..., T - J + 1\}$.

However, due to the high missing rate of the target observed data and historical data segments, their observed data positions may not fully align, which severely affects the calculation of similarity between datasets. To avoid inaccuracies in matching due to missing values, we calculate similarity only for the data points that are present in both the target observational data $X \in \mathbb{R}^{I \times J}$ and the historical data segments $H^t \in \mathbb{R}^{I \times J}$.

First, we identify their common sampling positions by calculating their intersection mask matrix as follows:

$$M^{\mathrm{inter}\_t} = M \cap M^t, \tag{2}$$

where $M$ and $M^t$ represent the mask matrices for the target observational data $X$ and the historical data segment $H^t$, respectively. The mask matrix $M^{\mathrm{inter}\_t}$ indicates which positions have observed values in both the target observational data $X$ and the historical data segment $H^t$. If both $X_{i,j}$ and $H_{i,j}$ have valid observations at position $(i, j)$, then $M_{i,j}^{\mathrm{inter}\_t} = 1$; otherwise, $M_{i,j}^{\mathrm{inter}\_t} = 0$.

Additionally, since the missing patterns in the historical data are completely random, the amount of observational data in the matched historical segments varies, impacting the fairness of similarity comparisons. To address this, we compute the mean Frobenius norm distance based on the available data in both the target observational data and the historical data segments as follows:

$$f_t = \frac{\sum_i^I \sum_j^J M_{i,j}^{\text{inter}\_t} \sqrt{\left(X_{i,j} - H_{i,j}^t\right)^2}}{\left\|M^{\text{inter}\_t}\right\|_1},$$ (3)

where $\left\|M^{\text{inter}\_t}\right\|_1$ denotes the number of intersecting elements between $X$ and $H^t$.

Note that a smaller mean Frobenius norm distance $f_t$ indicates a higher similarity between the historical data segment $H^t$ and the target observational data $X$. By comparing the mean Frobenius norm distance $f_t$ of $T - J + 1$ historical data segments $H^t$ with the target observational data $X$, we identify the top $K$ most similar historical data segments $[H^{t_1}, \ldots, H^{t_k}, \ldots, H^{t_K}]$ for each sample, where $k = \{1, 2, ..., K\}$.

Then, we design a *Fusion Process* to fuse the selected $K$ best match historical data segments with the training data $X_{\text{train}}$ more accurately.

**Fusion Process:** In the *Fusion Process*, we need to consider the following two factors:

- **Distance similarity.** The distance similarity between each selected historical data segment and the current target observational data varies. We believe that historical data segments with smaller mean Frobenius norm distance $f_t$ should be more important and apply higher weight in the *Fusion Process*.

- **Temporal proximity.** The closer the starting time slot of a historical data segment is to the current time slot, the more significant its data becomes. Therefore, we analyze the weight that should be assigned to each historical data segment during the *Fusion Process*, considering both distance similarity and temporal proximity.

Therefore, we design the *Fusion Process* based on the similarity and temporal proximity between the match historical data segments $H^t$ and current target observational data $X$.

*Distance similarity:* To further measure the similarity between the selected $K$ historical data segments $H^t$ and the target observational data $X$, we define a distance similarity score $s_k$ as follows:

$$s_k = 1 - \frac{f_{t_k}}{f_{t_1} + f_{t_2} + \cdots + f_{t_K}},$$ (4)

where $s_k$ reflects how closely the historical data segment matches the target observational data, the larger $s_k$ means the $k$-th selected historical data segments $H^{t_k}$ is more similar to the current target observational data $X$.

*Temporal proximity:* To measure temporal proximity between the selected $K$ historical data segments $H^t$ and the target observational data $X$, we define a temporal distance score $d_k$ as follows:

$$d_k = t_k/T,$$ (5)

where $t_k$ is the start time slot of the $k$-th selected historical data segments $H^{t_k}$, and $T$ is the latest time slot (i.e., the largest time slot in the historical data stream). The larger $d_k$ means the $k$-th selected historical data segments $H^{t_k}$ is closer to the current target observational data $X$.

*Fusion weight:* To consider both the similarity and temporal proximity between the match historical data segments $H^t$ and current target observational data $X$, we defined the weight $\lambda_k$ as follows:

$$\lambda_k = \frac{\mu d_k + (1 - \mu)s_k}{\mu \sum_{i=1}^K d_i + (1 - \mu) \sum_{j=1}^K s_j},$$ (6)

where $\mu \in (0, 1)$ is a learnable hyperparameter used to adjust the weight between the similarity score $s_k$ and the temporal distance score $d_k$. The larger $\lambda_k$ means the $k$-th selected historical data segments $H^{t_k}$ is more similar and closer to the current target observational data $X$, thus we need to apply larger weight to this historical data segment in the *Fusion Process*.

*Fusion process:* Since our goal is to supplement the training data $X_{\text{train}}$ with historical data segments, we tend to trust the data from $X_{\text{train}}$ rather than the data from historical data segments. Moreover, to avoid overwriting the supervised data $X_{\text{sup}}$, we need to ensure the locations where $X_{\text{sup}}$ has observational data are set to zero in fused data. Thus, we formulate the *Fusion Process* as follows:

$$X_{i,j}^{\text{fu}} = \begin{cases} X_{\text{train}(i,j)}, & X_{\text{train}(i,j)} \text{ is observed;} \\ 0, & X_{\text{sup}(i,j)} \text{ is observed;} \\ \frac{\lambda_n}{\lambda_n + ... + \lambda_m} H_{i,j}^{t_n} + ... + \frac{\lambda_m}{\lambda_n + ... + \lambda_m} H_{i,j}^{t_m}, & \text{otherwise,} \end{cases}$$ (7)

where $X_{\text{train}(i,j)}$ and $X_{\text{sup}(i,j)}$ denote the values at position $(i,j)$ in the training data $X_{\text{train}}$ and the supervised data $X_{\text{sup}}$, respectively. The set $\{H_{i,j}^{t_n}, \ldots, H_{i,j}^{t_m}\}$ includes the historical data segments among the $K$ matched segments that contain observed values at position $(i,j)$.

Noted that, in Eq.(7), the $\frac{\lambda_n}{\lambda_n + \ldots + \lambda_m} + \ldots + \frac{\lambda_m}{\lambda_n + \ldots + \lambda_m} = 1$, ensuring that the data remains within a reasonable range after fusion.

We present fusion examples in Appendix 7.2.

Based on our proposed historical data supplement module, our training data first identifies the most matching historical data segments through a matching process. Next, through a well-designed *Fusion Process*, the matched historical data segments are effectively fused with $X_{\text{train}}$. This process ensures that our training data is effectively supplemented with historical data, providing ample training data for the subsequent imputation module.

### 4.2 DIFFUSION IMPUTATION MODULE

The HDDI model proposed in this paper primarily relies on the diffusion model for data imputation. As illustrated in Fig.7, the HDDI model comprises two main stages: the diffusion process (training) and the denoising process (imputation).

In the diffusion process, fused training data $X^{\text{fu}}$ and supervised data $X_{\text{sup}}$ are used as inputs. We progressively add Gaussian noise into $X_{\text{sup}}$, eventually transforming it into pure noise $X_{\text{sup}}^N$. During this process, we train a noise estimation model $\epsilon_\theta$ by comparing the Gaussian noise $\epsilon$ added to the supervised data at each step with the noise $\epsilon_\theta(X_{\text{sup}}^n, X^{\text{fu}}, n)$ estimated by $\epsilon_\theta$.

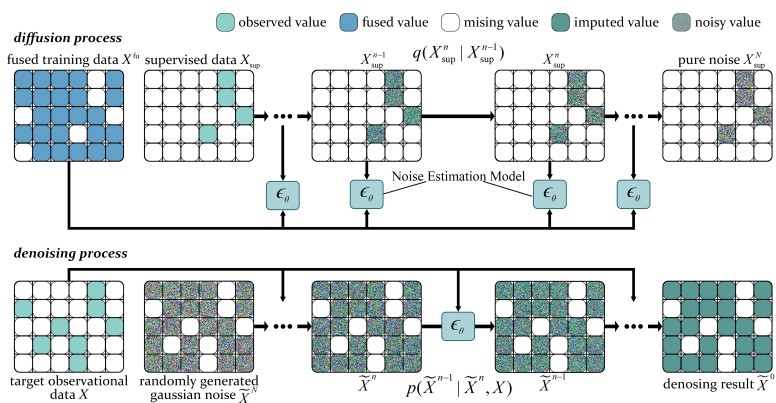

Figure 7: Visualization process of Diffusion Imputation Module.

The denoising process begins with initializing random Gaussian noise $\widetilde{X}^N$ and target observational data $X$, which serves as input. We estimate the noise at each step using the noise estimation model $\epsilon_\theta$ and progressively remove the noise from $\widetilde{X}^N$. Based on it, we can transforms $\widetilde{X}^N$ into the denoising result $\widetilde{X}^0$.

Finally, we can obtain the imputed result $X^{\text{result}}$ based on the target observational data $X$ and denoising result $\widetilde{X}^0$.

**Diffusion Process:** The diffusion process aims to train the noise estimation model $\epsilon_\theta$. During this process, noise is progressively added to the target observational data, gradually transforming it into pure noise. The optimization of the noise estimation model is achieved by minimizing the discrepancy between the noise added at each step and the noise estimated by the model. The specific process is detailed below.

We add the Gaussian noise $\epsilon$ into the supervised data $X_{\text{sup}}$ incrementally to produce noisy data $X_{\text{sup}}^n$ at each step, where $n = 1, 2, ..., N$ as follows:

$$X_{\text{sup}}^n = \sqrt{1 - \beta_n} X_{\text{sup}}^{n-1} + \beta_n \epsilon, \tag{8}$$

where $\epsilon \sim \mathcal{N}(0, 1)$ and $\beta_n \in (0, 1)$ denotes the level of noise.

Noted that, when $n$ is small, a smaller $\beta_n$ ensures that less Gaussian noise is added to $X_{\text{sup}}^{n-1}$ at step $n$. With the increase of $n$, the $\beta_n$ gradually increases. Thus, in the diffusion process, $X_{\text{sup}}$ undergoes $N$ steps of Gaussian noise addition to eventually result in a pure noise $X_{\text{sup}}^N$.

Since the process of adding noise is iterative after we unfold Eq.(8), we derive the formula directly from $X_{\text{sup}}$ to obtain $X_{\text{sup}}^n$:

$$X_{\text{sup}}^n = \sqrt{\widetilde{\alpha}_n} X_{\text{sup}} + \sqrt{1 - \widetilde{\alpha}_n} \epsilon, \tag{9}$$

where $\widetilde{\alpha}_n := \Pi_{i=1}^n \alpha_i$ and $\alpha_i := 1 - \beta_i$.

Since we input the noisy supervised data $X_{\text{sup}}^n$ in the $n$-th iteration and the fused training data $X^{\text{fu}}$ into the model, we can obtain the corresponding estimated noise: $\epsilon_\theta(X_{\text{sup}}^n, X^{\text{fu}}, n)$.

Then we can train the noise estimation model $\epsilon_\theta$ by minimizing the loss between the Gaussian noise $\epsilon$ added at each step and the noise $\epsilon_\theta$ estimated by the model as follows:

$$\min_\theta L(\theta) := \min_\theta \mathbb{E}_{X_{\text{sup}},\epsilon,n}||\epsilon - \epsilon_\theta(X_{\text{sup}}^n, X^{\text{fu}}, n)||_2^2, \qquad (10)$$

where $\theta$ is the trainable parameters in the noise estimation model $\epsilon_\theta$.

We organize the entire diffusion process as Algorithm 1. First, we randomly sample an $n$ from the set total diffusion steps $N$ as the diffusion step for the current iteration. As shown in Fig.8, during the process of adding noise in any arbitrary $n$-th iteration, we calculate the noisy supervised data $X_{\text{sup}}^n$ based on Eq.(9) and input it along with the fused training data $X^{\text{fu}}$ into the noise estimation model to obtain the estimated noise results $\epsilon_\theta(X_{\text{sup}}^n, X^{\text{fu}}, n)$. Next, we can train the noise estimation model by minimizing the loss function Eq.(10). Iterating this process until the model converges.

---

**Algorithm 1** Diffusion Process

---

**Input:** Supervised data $X_{\text{sup}}$, fused training data $X^{\text{fu}}$, the number of diffusion step $N$.
**Output:** Noise estimation model $\epsilon_\theta$.
1: **repeat**
2:   Sample the diffusion step $n \sim \text{Uniform}(\{1, ..., N\})$.
3:   Generate the Gaussian noise $\epsilon \sim \mathcal{N}(0, 1)$.
4:   Calculate noisy data $X_{\text{sup}}^n$ based on Eq.(9).
5:   Train the noise estimation model based on Eq.(10).
6: **until** converged

---

**Denoising process:** With the noise estimation model $\epsilon_\theta$ trained in the previous section, we now initialize Gaussian noise data $\widetilde{X}^N$ randomly and use target observational data as conditional data for the noise estimation model. By inputting the randomly generated Gaussian noise $\widetilde{X}^N$ along with the noise estimation model $\epsilon_\theta$, we perform denoising to obtain the denoised result $\widetilde{X}^0$. The specific process is as follows:

The denoising process consists of $N$ steps. As shown in Fig.9, the $n$-th step of the denoising process involves the following two steps:

- Step 1: Input the target observational data $X$ and the denoised noisy data $\widetilde{X}^n$ in the $n$-th iteration, estimate the noise based on the noise estimation model, which can be denoted as $\epsilon_\theta(\widetilde{X}^n, X, n)$.

- Step 2: Remove the noise estimated by the noise estimation model $\epsilon_\theta(\widetilde{X}^n, X, n)$ from the noisy data $\widetilde{X}^n$ to obtain $\widetilde{X}^{n-1}$, which represents the noisy data at step $n-1$.

The denoising process in the $n$-th iteration can be represented as:

$$\widetilde{X}^{n-1} = \frac{1}{\sqrt{\alpha_n}}\left(\widetilde{X}^n - \frac{\beta_n}{\sqrt{1-\widetilde{\alpha}_n}}\epsilon_\theta(\widetilde{X}^n, X, n)\right) + \sigma_n^2\mathbf{I}, \qquad (11)$$

where $\mathbf{I} \sim \mathcal{N}(0, 1)$, and $\sigma_n^2$ is defined as follows:

$$\sigma_n^2 = \begin{cases} \frac{1-\widetilde{\alpha}_{n-1}}{1-\widetilde{\alpha}_n}\beta_n, & \text{for } n > 1; \\ \beta_1, & \text{for } n = 1. \end{cases} \qquad (12)$$

We organize the entire denoising process as Algorithm 2. The imputation process starts by denoising from a random Gaussian noise $\widetilde{X}^N$, with the number of steps $n$ decreasing from $N$ to 1. In the $n$-th iteration of the step-by-step imputation process, the noise estimation model utilizes the current noisy data $\widetilde{X}^n$ and target observational data $n$ to estimate the noise $\epsilon_\theta(\widetilde{X}^n, X, n)$ for the current step. Subsequently, the next noisy data $\widetilde{X}^{n-1}$ for the next step is computed using Eq.(11) and Eq.(12). This process iterates until $n = 1$ to obtain the imputation result $\widetilde{X}^0$.

$$X^{\text{result}} = M \odot X + (1 - M) \odot \widetilde{X}^0, \qquad (13)$$

where $M$ represents the mask of the target observational data $X$.

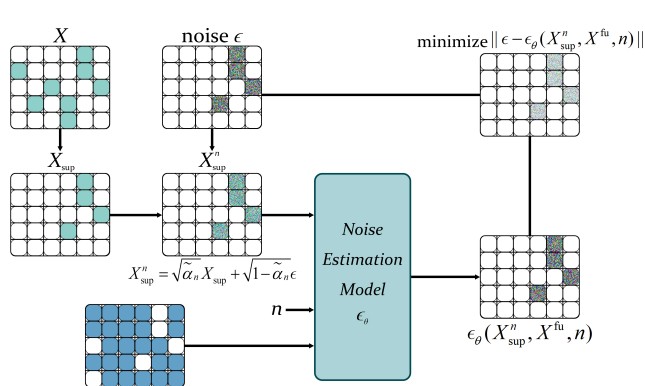

Figure 8: The $n$-th step of the diffusion process.

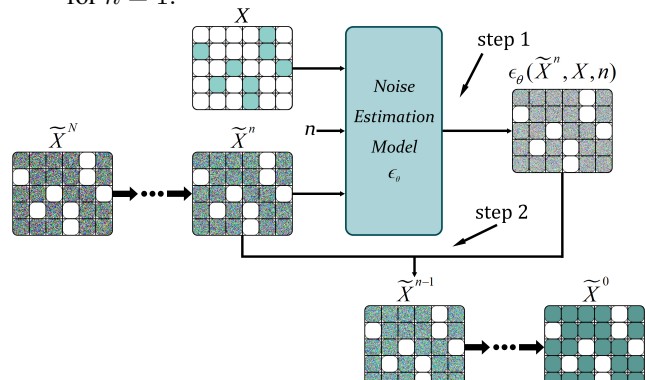

Figure 9: The $n$-th step of the denoising process.

**Imputation process:** After we obtain the final denoised result by de denoising process, we fuse the denoised result $\widetilde{X}^0$ with the target observational data $X$ to obtain the imputation result $X^{\text{result}}$:

---

**Algorithm 2** Denoising Process

---

**Input:** Target observational data $X$, the number of diffusion step $N$, randomly initialize Gaussian noise data $\widetilde{X}^N \sim \mathcal{N}(0,1)$ where the dimension of $\widetilde{X}^N$ corresponds to $X$.

**Output:** Denoising result $\widetilde{X}^0$.

1: **for** $n = N, ..., 1$ **do**
2: $\quad$ $\mathbf{I} \sim \mathcal{N}(0,1)$ if $n > 1$ else $\mathbf{I} = 0$.
3: $\quad$ Compute the estimated noise $\epsilon_\theta(\widetilde{X}^n, X, n)$ using the noise estimation model $\epsilon_\theta$.
4: $\quad$ Calculate the denoising result $\widetilde{X}^{n-1}$ according to Eq.(11) and Eq.(12).
5: **end for**
6: **return** denoising result $\widetilde{X}^0$.

---

**Algorithm 3** HDDI

---

**Input:** target observational data $\mathcal{X}$, historical data streams $\mathcal{X}^{\text{hist}}$, the number of matched history data $K$, the number of diffusion step $N$.

**Output:** Imputation results $\mathcal{X}^{\text{result}}$.

1: Search the top-$K$ matching history data $[H^{t_1}, ..., H^{t_k}, ..., H^{t_K}]$ from $\mathcal{X}^{\text{hist}}$ by *Historical Data Matching Process*.
2: Fuse the training data $X_{\text{train}}$ and $[H^{t_1}, ..., H^{t_k}, ..., H^{t_K}]$ using Eq.(7), to obtain the fused training data $\mathcal{X}^{\text{fu}}$.
3: Train the noise estimation model $\epsilon_\theta$ by Algorithm 1.
4: Generate the denoising result $\widetilde{X}^0$ by Algorithm 2.
5: Calculate imputation results

$$\mathcal{X}^{\text{result}} = M \odot \mathcal{X} + (1 - M) \odot \widetilde{X}^0.$$

6: **return** Imputation results $\mathcal{X}^{\text{result}}$.

---

Finally, our HDDI can be organized in Algorithm 3. We first select the $K$ historical data segments most similar to the current target observational data $X$ based on the *Historical Data Matching Process*. Then, using the *Fusion Process*, we fuse these historical data segments with the training data $X_{\text{train}}$ in appropriate proportions to obtain the fused training data $X^{\text{fu}}$, which is supplemented with historical data.

Next, the supervised data $X_{\text{sup}}$ and the fused training data $X^{\text{fu}}$ are input into the diffusion imputation module for training the noise estimation model $\epsilon_\theta$. Using this noise estimation model, we progressively remove noise from pure Gaussian noise $\widetilde{X}^N$ to generate the denoised result $\widetilde{X}^0$, thereby obtaining the final imputation result $\mathcal{X}^{\text{result}}$.

In summary, utilizing the fused training data $\mathcal{X}^{\text{fu}}$ obtained from the historical data supplement module can provide more comprehensive training data for model training, leading to improved imputation accuracy in both high missing rate and long-term missing data scenarios, as demonstrated in the experimental section.

## 5 EXPERIMENT

In this section, we evaluate our proposed HDDI by comparing it with five state-of-the-art baseline methods under five multivariate time series datasets. The dataset, baseline methods, and evaluation metrics used in the experiment are detailed in Appendix 7.3. In addition, we simulated two data missing scenarios (including random missing and long-term missing) to validate the robustness and effectiveness of our algorithm. In summary, we will investigate the following questions through experiments:

**RQ1:** Can our HDDI effectively handle the high missing rate? **RQ2:** If our HDDI is effective in the long-term missing data scenario? **RQ3:** Can our HDDI achieve better performance compared to existing state-of-the-art (SOTA) methods? **RQ4:** Is the *Historical Data Matching Process* effective? **RQ5:** Is the *Fusion Process* effective?

### 5.1 COMPARED WITH OTHER SOTA BASELINE METHODS

To address **RQ1**, we conducted experiments under the RM scenario to demonstrate the effectiveness of our imputation algorithm in high missing rate scenarios. As shown in Table 2 of Appendix 7.4, by comparing with five SOTA baseline methods, we can observe:

**With the increase of missing rate, the imputation accuracy of all methods decreases.** As the missing rate increases, the training data gradually becomes insufficient, leading to a decrease in imputation accuracy.

**Compared with other SOTA baseline methods, our method achieves the highest accuracy across all missing rates in random missing scenarios.** With the data missing rate increases, the gap between our proposed method and other baselines increases, because our method supplements training data with similar data from historical records, addressing the issue of insufficient training data caused by a low sampling rate.

To address **RQ2** and validate the effectiveness of our imputation algorithm in LM scenarios, we compare our HDDI with five baseline methods. As shown in Table 3 of Appendix 7.4, we can observe:

**With the increase of missing rate, the imputation accuracy of all methods decreases.** As the missing rate increases, the training data gradually becomes insufficient, leading to a decrease in imputation accuracy.

**Compared with other SOTA baseline methods, our HDDI achieves the highest accuracy across all missing rates in long-term missing data scenarios.** Since our HDDI approach uses historical data segments similar to the target observational data to supplement the training data, the long-term missing data segments are supplemented by historical

data segments. As a result, the model can resolve the underfitting problem associated with undamaged equipment, thereby enhancing the accuracy of missing data imputation.

Based on the results of the comparative experiments on the two missing scenarios described above, we can answer **RQ3**: Our imputation scheme achieves higher accuracy and greater reliability compared to existing SOTA methods. This demonstrates that our HDDI approach effectively identifies historical data most similar to the current target observational data, thereby providing sufficient and reliable training data. This capability enables our model to handle diverse and complex time series missing scenarios, including both random and long-term missing.

Even in scenarios with high missing rate data, our HDDI still achieves excellent imputation performance. For instance, when the data missing rate reaches 90%, HDDI achieves an average MAE of 0.3758 across five datasets in the random missing scenario, which represents a 25.15% improvement in accuracy compared to the best baseline method (CSDI). In long-term missing scenarios, HDDI has an average MAE of 0.4635, which is a 13.64% accuracy improvement over the best baseline method (GP-VAE).

## 5.2 ABLATION STUDIES

To prove the effectiveness of the *Historical Data Matching Process* and the *Fusion Process* in our HDDI, we conduct ablation studies as follows:

Firstly, to address **RQ4** and validate the effectiveness of the *Historical Data Matching Process*, we randomly select 10%, 30%, 50%, 70%, and 90% of observations as missing values from two datasets: ETT and Air-Quality. We design three sets of experiments for comparison:

1) The first set uses the nearest historical data segments as the data source for fusion from the history data stream, which is denoted as HDDI(near-match).

2) The second set randomly selects historical data segments as the data source for fusion from the history data stream, which is denoted as HDDI(rand-match).

3) The third set does not use data from the history data stream as a supplement, instead directly training the diffusion process with only target observational data, which is denoted as HDDI(no-history).

After the *Historical Data Matching Process*, we proceed with the remaining steps as described in this paper to evaluate the imputation results. As shown in Table 4 of Appendix 7.4, we can observe:

**Our designed historical data matching scheme achieves better performance compared to all other three matching methods.** Due to the *Historical Data Matching Process* we designed, which finds historical data most similar to the current target observation data using a sliding window-based approach, the supplemented historical data has features closer to the current imputation data. This ensures that they provide more accurate information for the diffusion imputation module and achieve higher imputation performance.

To address **RQ5** and validate the effectiveness of the *Fusion Process*, we use the same random missing scenario on the ETT and Air Quality datasets with 10%, 30%, 50%, 70%, and 90% missing rates. We conduct a comparison experiment called HDDI(aver-fusion), where, after identifying historical data segments similar to the current target observational data, the fusion stage does not consider the similarity and temporal correlation scores. Instead, it uses a simple averaging method to fuse the data, using the average of historical data as supplementary data for training the model. As shown in Table 5 of Appendix 7.4, we can observe:

**Our designed fusion scheme consistently outperforms the averaging fusion scheme across various missing rates.** Due to the design of our *Fusion Process*, which thoroughly considers the similarity in distance and temporal proximity between historical data and current target observational data, we place more trust in historical data that is closer in both time and distance to the current moment. As a result, our fused outcome is more accurate compared to simply taking the average value and achieves higher accuracy imputation.

These ablation studies highlight the significance of both the *Historical Data Matching Process* and the *Fusion Process* in our framework, demonstrating their contributions to achieving accurate and reliable imputation results.

## 6 CONCLUSION

To ensure high-precision imputation of missing data in multivariate time series under scenarios of high missing rate or long-term data absence, we propose the HDDI model. By designing a historical data supplementation module, we select multiple historical data segments which most similar to the current target observation data from the historical data stream and fuse them with the current training data. To fully utilize the supplemented training data, we design a diffusion imputation module to achieve high-accuracy imputation in scenarios of high missing rate and long-term missing. Extensive experimental results demonstrate that our method outperforms in both high missing rate and long-term missing scenarios across five multivariate time series datasets.

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

## 7 APPENDIX

### 7.1 NOTATIONS

We organize the meanings of the symbolic expressions in this paper in Table 1.

Table 1: Important notations for the HDDI Model

| Notations | Descriptions |
|---|---|
| $X$ | Target observational data |
| $M$ | Mask matrix for target observational data |
| $X^{\text{fu}}$ | Fused training data |
| $X^{\text{result}}$ | Imputation results |
| $H^t$ | Historical data segments |
| $H^{t_k}$ | Selected historical data segments |
| $M^{\text{inter}\_t}$ | Intersection mask matrix |
| $X_{\text{train}}$ | Training data (a portion of $X$) |
| $X_{\text{sup}}$ | Supervised data (a portion of $X$) |
| $X_{\text{sup}}^{1:N}$ | Results of each step in the diffusion process |
| $\widetilde{X}^N$ | Randomly generated Gaussian noise |
| $\widetilde{X}^{1:N-1}$ | Results of each step in the denoising process |
| $\widetilde{X}^0$ | Denoising result |
| $K$ | The number of selected historical data segments |
| $T$ | Length of historical data stream |
| $N$ | The total number of diffusion steps |

### 7.2 FUSION EXAMPLES

More specifically, as shown in Fig.6(b), we selected three historical data segments: $H^{t_1}$, $H^{t_2}$, and $H^{t_3}$. The *Fusion Process* can be divided into three scenarios:

1) Taking the $X_{3,3}^{\text{fu}}$ as an example, if position $(i, j)$ has observational data in the training data $X_{\text{train}}$, the corresponding position in the fused training data can be represented as $X_{3,3}^{\text{fu}} = X_{\text{train}(3,3)}$.

2) Taking the $X_{4,4}^{\text{fu}}$ as an example, if position $(i, j)$ has observational data in the supervised data $X_{\text{sup}}$, the corresponding position in the fused training data can be represented as $X_{4,4}^{\text{fu}} = 0$.

3) Taking the $X_{1,1}^{\text{fu}}$ as an example, if position $(i, j)$ only has observational data in the historical data segments $H^{t_1}$ and $H^{t_2}$, the corresponding position in the fused training data can be represented as $X_{1,1}^{\text{fu}} = \frac{\lambda_1}{\lambda_1+\lambda_2} H_{1,1}^{t_1} + \frac{\lambda_2}{\lambda_1+\lambda_2} H_{1,1}^{t_2}$.

### 7.3 DATASET, BASELINE METHODS, AND EVALUATION METRICS

**Datasets:** *Electricity Transformer Temperature (ETT)* Zhou et al. (2021): The ETT dataset, collected from power transformers from July 1, 2016, to June 26, 2018, contains 69,680 samples, each featuring seven features, including oil temperature and six types of external power load characteristics. We use the last 2 hours as target observational data, while the first 22 hours are used as historical data.

*Beijing Multi-Site Air-Quality (Air-Quality)* Yi et al. (2016): The Air-Quality dataset includes PM2.5 measurements from 36 stations in Beijing, sampled hourly over 12 months. Each sequence consists of 36 consecutive time steps. The dataset itself contains approximately 13% missing data. We use the last 6 hours as target observational data, while the first 42 hours are used as historical data.

*Chicago Crime Dataset (Chicago-Crime)* Aldossari et al. (2020): The Chicago-Crime dataset spans from January 1, 2001, to December 11, 2017. It provides information on crime time, location, and type. We process the dataset into a three-dimensional structure with 32 crime types, 200 months, and 77 locations. We use the last 10 months to serve as target observational data, while the first 180 months are used as historical data.

*Electricity Load Diagrams (Electricity)* Lai et al. (2018): The Electricity dataset, sourced from UCI. It contains electricity consumption data collected every 15 minutes from 370 customers over a period from January 1, 2011, to December 31, 2014 (48 months). We use the last 1 hour as target observational data, while the first 23 hours are used as historical data.

*PhysioNet Challenge 2012 (PhysioNet-2012)* Goldberger et al. (2000): The PhysioNet-2012 dataset includes 4,000 clinical time series from intensive care units (ICUs) over 48 hours, with 35 variables. The dataset itself contains approximately 80% missing data. We use the last 6 hours as target observational data, while the first 42 hours are used as historical data.

**Baseline Methods:** We implement five SOTA deep learning-based multivariate series imputation methods as follows:

- BRITS Cao et al. (2018): Based on the recurrent neural network model, BRITS uses bidirectional recurrent neural networks to fuse bidirectional information flow, learning patterns from the context around missing data for imputation.
- GP-VAE Fortuin et al. (2020): Based on the variational autoencoder model, GP-VAE utilizes a deep variational autoencoder (VAE) to obtain latent representations and employs the Gaussian process in latent space to capture the global dynamics and structure of time series.
- SS-GAN Miao et al. (2021): Combining RNN with a semi-supervised generative adversarial network model, SS-GAN uses semi-supervised learning mechanisms to guide the learning process of generators and discriminators.
- SAITS Du et al. (2023): A model based on a self-attention mechanism, SAITS models dependencies between sequence elements directly using self-attention mechanisms.
- CSDI Tashiro et al. (2021): Uses a conditional score-based diffusion model to estimate missing values by gradually transforming noise into coherent time series.

**Evaluation Metrics:** We evaluate each method using the following metrics:

$$\text{MAE} = (\sum_{i}^{I} \sum_{j}^{J} |(X^{\text{result}} - X) \odot M^{\text{test}}|)/\left\| M^{\text{test}} \right\|_1, \tag{14}$$

$$\text{RMSE} = \sqrt{(\sum_{i}^{I} \sum_{j}^{J} ((X^{\text{result}} - X) \odot M^{\text{test}})^2)/\|M^{\text{test}}\|_1}. \tag{15}$$

- **Mean Absolute Error (MAE):** Measures the average absolute difference between predicted value and ground-truth values of missing data, providing an intuitive and interpretable error measure.
- **Root Mean Square Error (RMSE):** Measures the square root of the average squared difference between predicted and actual values. It is sensitive to larger errors, thus indicating significant deviations between predictions and actual values.

The mathematical definition of the evaluation metric is as follows. Note that we set the mask for the test data as $M^{\text{test}}$, while $\|M^{\text{test}}\|_1$ represents the number of test data.

**Missing Data Scenarios:**

Our experiment simulated two common missing data scenarios: Random Missing and Long-term Missing.

- **Random Missing (RM):** We randomly select $\gamma\%$ of the data as missing data, while the remaining data serves as the target observational data $X$. To evaluate the accuracy of the imputation algorithm under different missing rates, we set $\gamma$ to 10, 30, 50, 70, and 90, respectively.
- **Long-term Missing (LM):** Due to the high level of missing data in the PhysioNet healthcare dataset, it is not feasible to approximate the long-term missing scenario. Instead, we select the other four datasets to test the long-term missing scenario. We randomly select $\gamma\%$ of the features from each sample to be damaged. For each damaged feature, we set all data points at all time slots to be missing. Similar to the random missing scenario, we set $\gamma$ to 10, 30, 50, 70, and 90, respectively.

The HDDI epoch is set to 1000, the learning rate is 0.001, the batch size is 16, and the number of diffusion steps is 50. All other baseline methods use their optimal parameters and are trained on a single NVIDIA GeForce RTX 3090 GPU. All experimental results are averaged over five trials.

## 7.4 EXPERIMENTAL RESULT

Table 2: Experimental Results in RM Scenarios.

| Dataset: ETT | | Missing Rate | | | | |
|---|---|---|---|---|---|---|
| Metrics | Model | 10% | 30% | 50% | 70% | 90% |
| MAE | BRITS | 0.2060 | 0.3889 | 0.5911 | 0.7278 | 0.7602 |
| | GP-VAE | 0.2325 | 0.3444 | 0.4882 | 0.6189 | 0.7237 |
| | SS-GAN | 0.2252 | 0.3641 | 0.5403 | 0.7229 | 0.7602 |
| | SAITS | 0.0984 | 0.1750 | 0.6285 | 0.7602 | 0.7644 |
| | CSDI | 0.0887 | 0.1003 | 0.1311 | 0.2237 | 0.4530 |
| | **HDDI** | **0.0718** | **0.0851** | **0.1035** | **0.1712** | **0.3609** |
| RMSE | BRITS | 0.3055 | 0.5352 | 0.7779 | 0.9180 | 0.9547 |
| | GP-VAE | 0.3391 | 0.4599 | 0.6392 | 0.7868 | 0.9122 |
| | SS-GAN | 0.3315 | 0.5022 | 0.7128 | 0.9144 | 0.9547 |
| | SAITS | 0.1563 | 0.2861 | 0.8154 | 0.9547 | 0.9607 |
| | CSDI | 0.1940 | 0.2041 | 0.2744 | 0.4590 | 0.6362 |
| | **HDDI** | **0.1230** | **0.1613** | **0.1731** | **0.2920** | **0.5371** |

| Dataset: Air-Quality | | Missing Rate | | | | |
|---|---|---|---|---|---|---|
| Metrics | Model | 10% | 30% | 50% | 70% | 90% |
| MAE | BRITS | 0.2643 | 0.3347 | 0.4759 | 0.6204 | 0.6508 |
| | GP-VAE | 0.2803 | 0.3596 | 0.4472 | 0.5473 | 0.6283 |
| | SS-GAN | 0.2590 | 0.3281 | 0.4212 | 0.5270 | 0.6259 |
| | SAITS | 0.1696 | 0.2792 | 0.4256 | 0.5697 | 0.6654 |
| | CSDI | 0.0920 | 0.1033 | 0.1272 | 0.1936 | 0.4907 |
| | **HDDI** | **0.0887** | **0.0989** | **0.1186** | **0.1776** | **0.3022** |
| RMSE | BRITS | 0.4180 | 0.4720 | 0.6303 | 0.7403 | 0.7832 |
| | GP-VAE | 0.4218 | 0.4823 | 0.5636 | 0.6641 | 0.7609 |
| | SS-GAN | 0.4118 | 0.4720 | 0.5661 | 0.6516 | 0.7664 |
| | SAITS | 0.3115 | 0.4153 | 0.5724 | 0.7077 | 0.8048 |
| | CSDI | 0.1614 | 0.1772 | 0.2248 | 0.3394 | 0.6392 |
| | **HDDI** | **0.1536** | **0.1650** | **0.2066** | **0.2974** | **0.4965** |

| Dataset: PhysioNet-2012 | | Missing Rate | | | | |
|---|---|---|---|---|---|---|
| Metrics | Model | 10% | 30% | 50% | 70% | 90% |
| MAE | BRITS | 0.3102 | 0.4191 | 0.6530 | 0.6961 | 0.6984 |
| | GP-VAE | 0.4313 | 0.4916 | 0.5590 | 0.6356 | 0.6967 |
| | SS-GAN | 0.3306 | 0.4429 | 0.6169 | 0.6961 | 0.6974 |
| | SAITS | 0.2512 | 0.3746 | 0.6961 | 0.6962 | 0.6972 |
| | CSDI | 0.2617 | 0.3277 | 0.4078 | 0.5532 | 0.6415 |
| | **HDDI** | **0.2407** | **0.2955** | **0.3617** | **0.4602** | **0.5936** |
| RMSE | BRITS | 0.5582 | 0.7096 | 0.9202 | 0.9675 | 0.9826 |
| | GP-VAE | 0.6560 | 0.7704 | 0.8192 | 0.8950 | 0.9608 |
| | SS-GAN | 0.5709 | 0.7290 | 0.8799 | 0.9674 | 0.9815 |
| | SAITS | 0.4965 | 0.6776 | 0.9676 | 0.9701 | 0.9716 |
| | CSDI | 0.5314 | 0.5623 | 0.7760 | 0.8363 | 0.9245 |
| | **HDDI** | **0.4693** | **0.5500** | **0.6021** | **0.7234** | **0.8531** |

| Dataset: Chicago-Crime | | Missing Rate | | | | |
|---|---|---|---|---|---|---|
| Metrics | Model | 10% | 30% | 50% | 70% | 90% |
| MAE | BRITS | 0.2613 | 0.3351 | 0.4115 | 0.5220 | 0.5422 |
| | GP-VAE | 0.3071 | 0.3490 | 0.4063 | 0.4558 | 0.5179 |
| | SS-GAN | 0.2697 | 0.3485 | 0.4236 | 0.5069 | 0.5417 |
| | SAITS | 0.2461 | 0.2964 | 0.3728 | 0.5192 | 0.5425 |
| | CSDI | 0.1994 | 0.2128 | 0.2277 | 0.2855 | 0.4526 |
| | **HDDI** | **0.1857** | **0.1910** | **0.2275** | **0.2700** | **0.3014** |
| RMSE | BRITS | 0.7870 | 0.8863 | 0.9901 | 0.9672 | 0.9576 |
| | GP-VAE | 0.9080 | 0.9338 | 0.9818 | 0.9341 | 0.9370 |
| | SS-GAN | 0.7769 | 0.8849 | 0.9874 | 0.9519 | 0.9442 |
| | SAITS | 0.7973 | 0.8837 | 0.9865 | 0.9738 | 0.9566 |
| | CSDI | 0.3583 | 0.4944 | 0.4873 | 0.5152 | 0.7035 |
| | **HDDI** | **0.3483** | **0.3515** | **0.4228** | **0.5051** | **0.5366** |

| Dataset: Electricity | | Missing Rate | | | | |
|---|---|---|---|---|---|---|
| Metrics | Model | 10% | 30% | 50% | 70% | 90% |
| MAE | BRITS | 0.1468 | 0.2236 | 0.4752 | 0.7979 | 0.8079 |
| | GP-VAE | 0.1787 | 0.2999 | 0.4427 | 0.5876 | 0.7347 |
| | SS-GAN | 0.1427 | 0.2590 | 0.4747 | 0.7942 | 0.7976 |
| | SAITS | 0.1273 | 0.2086 | 0.5106 | 0.7947 | 0.8004 |
| | CSDI | 0.1174 | 0.1290 | 0.1497 | 0.2247 | 0.4060 |
| | **HDDI** | **0.1109** | **0.1215** | **0.1461** | **0.1835** | **0.3209** |
| RMSE | BRITS | 0.2626 | 0.3487 | 0.6284 | 0.9417 | 0.9418 |
| | GP-VAE | 0.2670 | 0.3933 | 0.5481 | 0.7064 | 0.8711 |
| | SS-GAN | 0.2382 | 0.3766 | 0.6280 | 0.9380 | 0.9416 |
| | SAITS | 0.2282 | 0.3375 | 0.6551 | 0.9381 | 0.9446 |
| | CSDI | 0.1913 | 0.2106 | 0.2491 | 0.3666 | 0.5485 |
| | **HDDI** | **0.1842** | **0.2044** | **0.2479** | **0.3005** | **0.5030** |

Table 3: Experimental Results in LM Scenarios.

| Dataset: ETT | | Missing Rate | | | | |
|---|---|---|---|---|---|---|
| **Metrics** | **Model** | **10%** | **30%** | **50%** | **70%** | **90%** |
| MAE | BRITS | 0.5146 | 0.7278 | 0.7327 | 0.7474 | 0.7692 |
| | GP-VAE | 0.1905 | 0.3717 | 0.4268 | 0.4816 | 0.5905 |
| | SS-GAN | 0.5606 | 0.7594 | 0.7731 | 0.7288 | 0.7254 |
| | SAITS | 0.0986 | 0.3226 | 0.3478 | 0.3972 | 0.8258 |
| | CSDI | 0.1429 | 0.3481 | 0.4174 | 0.5881 | 0.8065 |
| | **HDDI** | **0.0970** | **0.2919** | **0.3074** | **0.3622** | **0.4810** |
| RMSE | BRITS | 0.5146 | 0.7278 | 0.7327 | 0.7474 | 0.7692 |
| | GP-VAE | 0.2596 | 0.3770 | 0.4746 | 0.4835 | 0.6980 |
| | SS-GAN | 0.8119 | 0.9466 | 0.9545 | 0.9584 | 0.9800 |
| | SAITS | 0.2236 | 0.4854 | 0.5315 | 0.5362 | 1.1157 |
| | CSDI | 0.2514 | 0.4749 | 0.5667 | 0.8636 | 1.0237 |
| | **HDDI** | **0.2052** | **0.4592** | **0.4634** | **0.5783** | **0.6361** |

| Dataset: Air-Quality | | Missing Rate | | | | |
|---|---|---|---|---|---|---|
| **Metrics** | **Model** | **10%** | **30%** | **50%** | **70%** | **90%** |
| MAE | BRITS | 0.3307 | 0.6221 | 0.6273 | 0.6368 | 0.6675 |
| | GP-VAE | 0.3007 | 0.3543 | 0.3677 | 0.4250 | 0.5473 |
| | SS-GAN | 0.4430 | 0.6245 | 0.6250 | 0.6414 | 0.6699 |
| | SAITS | 0.1831 | 0.3927 | 0.5482 | 0.6097 | 0.6924 |
| | CSDI | 0.1410 | 0.1954 | 0.4298 | 0.4744 | 0.6471 |
| | **HDDI** | **0.1382** | **0.1909** | **0.3075** | **0.3877** | **0.4511** |
| RMSE | BRITS | 0.4681 | 0.7976 | 0.7991 | 0.8137 | 0.8393 |
| | GP-VAE | 0.4403 | 0.5422 | 0.5437 | 0.6402 | 0.7820 |
| | SS-GAN | 0.6290 | 0.7980 | 0.8007 | 0.8129 | 0.8368 |
| | SAITS | 0.3112 | 0.6771 | 0.8572 | 0.9116 | 0.9159 |
| | CSDI | 0.2207 | 0.3261 | 0.6279 | 0.6678 | 0.8267 |
| | **HDDI** | **0.2192** | **0.3181** | **0.5005** | **0.6218** | **0.7490** |

| Dataset: Chicago-Crime | | Missing Rate | | | | |
|---|---|---|---|---|---|---|
| **Metrics** | **Model** | **10%** | **30%** | **50%** | **70%** | **90%** |
| MAE | BRITS | 0.3020 | 0.3823 | 0.4391 | 0.4501 | 0.4612 |
| | GP-VAE | 0.2853 | 0.2896 | 0.2914 | 0.3228 | 0.4135 |
| | SS-GAN | 0.2268 | 0.2371 | 0.2607 | 0.3147 | 0.3950 |
| | SAITS | 0.3476 | 0.4259 | 0.4332 | 0.4478 | 0.4559 |
| | CSDI | 0.1979 | 0.2284 | 0.3046 | 0.4334 | 0.4784 |
| | **HDDI** | **0.1904** | **0.2253** | **0.2302** | **0.2729** | **0.3581** |
| RMSE | BRITS | 0.8723 | 0.9590 | 0.9590 | 1.0144 | 1.0369 |
| | GP-VAE | 0.8990 | 0.9133 | 0.9292 | 0.9382 | 0.9974 |
| | SS-GAN | 0.7715 | 0.8675 | 0.8836 | 0.8902 | 0.9230 |
| | SAITS | 0.8881 | 0.9293 | 0.9587 | 1.0131 | 1.0362 |
| | CSDI | 0.3605 | 0.4905 | 0.7422 | 0.8517 | 0.9830 |
| | **HDDI** | **0.3359** | **0.3585** | **0.4088** | **0.4935** | **0.6809** |

| Dataset: Electricity | | Missing Rate | | | | |
|---|---|---|---|---|---|---|
| **Metrics** | **Model** | **10%** | **30%** | **50%** | **70%** | **90%** |
| MAE | BRITS | 0.2932 | 0.6841 | 0.7756 | 0.7710 | 0.7931 |
| | GP-VAE | 0.1661 | 0.1686 | 0.2127 | 0.4107 | 0.6200 |
| | SS-GAN | 0.5781 | 0.7065 | 0.7789 | 0.7860 | 0.7879 |
| | SAITS | 0.1576 | 0.2806 | 0.3286 | 0.3550 | 0.7794 |
| | CSDI | 0.1136 | 0.2413 | 0.2826 | 0.4129 | 0.7223 |
| | **HDDI** | **0.1073** | **0.1674** | **0.2033** | **0.2825** | **0.5636** |
| RMSE | BRITS | 0.3876 | 0.8520 | 0.9204 | 0.9347 | 0.9365 |
| | GP-VAE | 0.2474 | 0.2719 | 0.3309 | 0.4103 | 0.8294 |
| | SS-GAN | 0.7244 | 0.8655 | 0.9315 | 0.9320 | 0.9350 |
| | SAITS | 0.2454 | 0.3994 | 0.4573 | 0.5237 | 0.9684 |
| | CSDI | 0.1781 | 0.3484 | 0.3692 | 0.5418 | 0.8820 |
| | **HDDI** | **0.1703** | **0.2658** | **0.3038** | **0.4045** | **0.7350** |

Table 4: Ablation studies of *Historical Data Matching Process*.

| Dataset: ETT | | Missing Rate | | | | |
|---|---|---|---|---|---|---|
| Metrics | Model | 10% | 30% | 50% | 70% | 90% |
| MAE | HDDI(near-match) | 0.0765 | 0.0905 | 0.1392 | 0.2118 | 0.4012 |
| | HDDI(rand-match) | 0.0735 | 0.0873 | 0.1058 | 0.1779 | 0.3858 |
| | HDDI(no-history) | 0.0887 | 0.1003 | 0.1311 | 0.2237 | 0.4530 |
| | **HDDI** | **0.0718** | **0.0851** | **0.1035** | **0.1712** | **0.3609** |
| RMSE | HDDI(near-match) | 0.1322 | 0.1684 | 0.2366 | 0.3400 | 0.6431 |
| | HDDI(rand-match) | 0.1282 | 0.1648 | 0.1822 | 0.3022 | 0.6200 |
| | HDDI(no-history) | 0.1940 | 0.2041 | 0.2744 | 0.4590 | 0.6362 |
| | **HDDI** | **0.1230** | **0.1613** | **0.1731** | **0.2920** | **0.5371** |
| Dataset: Air-Quality | | Missing Rate | | | | |
| Metrics | Model | 10% | 30% | 50% | 70% | 90% |
| MAE | HDDI(near-match) | 0.0899 | 0.1018 | 0.1288 | 0.1983 | 0.3457 |
| | HDDI(rand-match) | 0.0909 | 0.1019 | 0.1260 | 0.2034 | 0.3860 |
| | HDDI(no-history) | 0.0920 | 0.1033 | 0.1272 | 0.1936 | 0.4907 |
| | **HDDI** | **0.0887** | **0.0989** | **0.1186** | **0.1776** | **0.3022** |
| RMSE | HDDI(near-match) | 0.1555 | 0.1703 | 0.2274 | 0.3337 | 0.5407 |
| | HDDI(rand-match) | 0.1600 | 0.1723 | 0.2211 | 0.3767 | 0.6887 |
| | HDDI(no-history) | 0.1614 | 0.1772 | 0.2248 | 0.3394 | 0.6392 |
| | **HDDI** | **0.1536** | **0.1650** | **0.2066** | **0.2974** | **0.4965** |

Table 5: Ablation studies of *Fusion Process*.

| Dataset: ETT | | Missing Rate | | | | |
|---|---|---|---|---|---|---|
| Metrics | Model | 10% | 30% | 50% | 70% | 90% |
| MAE | HDDI(aver-fusion) | 0.0765 | 0.0905 | 0.1392 | 0.2118 | 0.4012 |
| | **HDDI** | **0.0718** | **0.0851** | **0.1035** | **0.1712** | **0.3609** |
| RMSE | HDDI(aver-fusion) | 0.1300 | 0.1684 | 0.1874 | 0.3049 | 0.6101 |
| | **HDDI** | **0.1230** | **0.1613** | **0.1731** | **0.2920** | **0.5371** |
| Dataset: Air-Quality | | Missing Rate | | | | |
| Metrics | Model | 10% | 30% | 50% | 70% | 90% |
| MAE | HDDI(aver-fusion) | 0.0912 | 0.1012 | 0.1205 | 0.1995 | 0.4596 |
| | **HDDI** | **0.0887** | **0.0989** | **0.1186** | **0.1776** | **0.3022** |
| RMSE | HDDI(aver-fusion) | 0.1623 | 0.1655 | 0.2107 | 0.3285 | 0.7859 |
| | **HDDI** | **0.1536** | **0.1650** | **0.2066** | **0.2974** | **0.4965** |

