# OpenReview forum: "HDDI: A Historical Data-Based Diffusion Imputation Method for High-Accuracy Recovery in Multivariate Time Series with High Missing Rate and Long-Term Gap"
_ICLR.cc/2025/Conference — ICLR 2025 Conference Withdrawn Submission_

### Official Review · Reviewer_4h4z · 2024-10-30

**Soundness:** 2
**Presentation:** 2
**Contribution:** 2
**Rating:** 3
**Confidence:** 4

**Summary:**

This paper focuses on time series imputation, specifically aiming to improve existing diffusion models for these techniques by introducing the "Historical Data Supplement Module." This module is intended to enhance imputation accuracy in cases where certain variables have been missing for extended periods.

**Strengths:**

- The approach of using historical data matches to enhance final time series imputation is noteworthy.

- The implementation effectively considers both similarity to current data and temporal distance when selecting historical data. The chosen metrics are appropriate, and the compression of this information into the $\lambda$ value is efficiently handled.

- Finally, treating the $\mu$ parameter as a learnable value, rather than as a tunable hyperparameter, stands out as an interesting and valuable contribution.

**Weaknesses:**

- The result tables should not be relegated to the supplementary material. Supplementary material is intended for additional information that may aid in contextualizing the paper or for presenting extra results. However, the core evidence supporting the paper’s claims should remain in the main content. According to the ICLR author guide: "Note that reviewers are encouraged, but not required to review supplementary material during the review process," so the results should be accessible without referring to that section.

- The results are presented in normalized form, which may obscure the practical benefits of this method in real terms. It would be beneficial to include unnormalized results for clearer interpretation.

- Some definitions in the paper may be incorrect or challenging to interpret; further details are provided in the questions.

- There is a lack of information regarding the scalability of this method.

**Questions:**

- The literature review appears fairly up-to-date, yet there is a notable absence regarding diffusion models for imputation: PriSTI [1]. Is there a particular reason for its omission?

- The paper discusses GRIN [2], which demonstrates the need for imputing by extracting spatio-temporal representations through GNNs. This approach appears beneficial in cases with extended missing data periods, including virtual missing experiments where imputations are made in scenarios where a node has failed across all training samples, showing notable improvements. PriSTI iterates on CSDI [3] by integrating GNNs, among other features, to leverage these advantages. Is there any reason GNNs were not considered in this paper? This method seems potentially beneficial to the proposed approach.

- The method of matching records from historical data is quite interesting, but it appears that this approach could scale linearly in complexity as new records are added in production. This raises the following questions: Do you have any analysis on how complexity grows under this scenario? Are there any constraints in place to mitigate this, such as limiting the maximum time window considered?

- Lines 60–63 mention that diffusion models can avoid the error accumulation problem of RNNs. However, this seems to be a minor error, as specified in [1, 3], and as noted in lines 117–121 of this paper, this issue is actually mitigated by self-attention methods incorporated within transformer layers. This confusion appears again in lines 47–50, where it seems to imply that error accumulation is linked to self-attention mechanisms. Is this correct, or is there a misinterpretation on my part?

- Another potential source of confusion is the use of the terms "forward process" and "denoising process." The diffusion process specifically refers to adding noise to a sample, while the denoising process is the mechanism for reconstructing the original sample from the noise, which is the primary task we aim to teach diffusion models. During training, both the diffusion process and the denoising task are involved. However, in lines 101–103, the text seems to imply that the forward process refers to model training, and that denoising corresponds to inference. Could this be a minor error, or is there a misinterpretation on my part?

- Finally, the approach of searching for historical values for missing data recalls PriSTI’s method of providing diffusion models with linearly interpolated imputed data to serve as conditional information [1]. Has the similarity been explored in this context?

[1] Liu, M., Huang, H., Feng, H., Sun, L., Du, B., & Fu, Y. (2023, April). Pristi: A conditional diffusion framework for spatiotemporal imputation. In 2023 IEEE 39th International Conference on Data Engineering (ICDE) (pp. 1927-1939). IEEE.

[2] Cini, A., Marisca, I., & Alippi, C. (2021). Filling the g_ap_s: Multivariate time series imputation by graph neural networks. arXiv preprint arXiv:2108.00298.

[3] Tashiro, Y., Song, J., Song, Y., & Ermon, S. (2021). Csdi: Conditional score-based diffusion models for probabilistic time series imputation. Advances in Neural Information Processing Systems, 34, 24804-24816.

---

### Official Review · Reviewer_ZUdh · 2024-10-31

**Soundness:** 2
**Presentation:** 2
**Contribution:** 2
**Rating:** 5
**Confidence:** 2

**Summary:**

The paper presents HDDI, a novel imputation method aimed at addressing high missing rates and long-term gaps in multivariate time series data. HDDI combines historical data supplementation with a diffusion-based imputation model to improve data recovery accuracy. By using historical data to fill gaps in the training dataset, HDDI overcomes limitations in deep learning models that typically struggle with insufficient data in high-missing-rate contexts. Additionally, HDDI employs a unique historical data fusion process, blending target observational data with matched historical data to enhance imputation precision. Finally, five experimental datasets are present to reveal HDDI’s significant accuracy improvements.

**Strengths:**

The paper introduces a unique method of imputation that integrates historical data in a diffusion-based model, providing a fresh perspective for handling high missing rates and long-term gaps. The method is evaluated against five baseline methods across multiple datasets, confirming its robustness and adaptability.
The authors explore both random and long-term missing scenarios, enhancing the practical applicability of HDDI for real-world time series imputation.
Each component of HDDI (historical data supplementation, diffusion imputation) is clearly explained and well-justified, showcasing a thoughtful and methodologically sound approach.

**Weaknesses:**

Please refer to questions section.

**Questions:**

1. In my opinion, the use of sliding windows for matching historical data may present significant computational challenges, especially with large datasets or those sampled at high frequencies. Could the authors discuss their approach to selecting the optimal sliding window length and how this affect the efficency of the HDDI?

2. Following the previous question, could the authors provide a detailed complexity analysis of the HDDI pipeline? This would be useful to understand real-time feasibility in large-scale data applications.

3. In Appendix 7.3, the baseline methods section considers only five methods, most of which are from pre-2021. Including a broader range of state-of-the-art methods would strengthen the comparison, for instance, TimesNet and iTransformer.

4. In Appendix 7.3's missing data scenarios section, the methodology for generating LM scenariosis not clear to me. The authors mention selecting damaged features and removing points from these features. Does this imply that remaining features are fully observed while all points in damaged features are removed?

---

### Official Review · Reviewer_Z1sb · 2024-11-03

**Soundness:** 2
**Presentation:** 2
**Contribution:** 2
**Rating:** 3
**Confidence:** 5

**Summary:**

This paper presents a solution to the time series imputation problem, particularly for datasets with high proportions of missing observed values. The proposed method, HDDI, comprises two distinct modules: the Historical Data Supplement Module, which selects analogous data windows from historical records to enrich the current training dataset, thereby mitigating the problem of insufficient training data due to a high missing rate. The Diffusion Imputation Module generates missing data based on a diffusion-based model through a process of adding and removing noise. Experimental validation shows that the method outperforms existing techniques, especially in scenarios with high missing data.

**Strengths:**

The time series imputation problem is important in the real world.

**Weaknesses:**

1. This version of the paper is not well-written. This work presents an overly detailed description of the background and methodology, consuming excessive space and leading to a sparse section on the experiments, which contains only limited analytical descriptions. Specific experimental results and even the definitions of symbols used in the methods are placed in the appendix. It is crucial to note that the appendix is not a section that readers are required to focus on; the omission of essential experimental results may confuse readers and undermine the credibility of the experimental conclusions. This significantly detracts from both the readability and the validity of the paper.

2. The approach of incorporating historical data during training is well-established and represents one of the standard methods in the field of time series analysis. Furthermore, diffusion models have been extensively employed in previous research on time series data imputation. The primary innovation of this paper lies in its utilization of historical data; however, aside from this aspect, there are insufficient novel contributions to substantiate the overall novelty of this paper.

3. The historical data matching method discussed in the paper relies on a sliding window with a step size of 1, which results in considerable time overhead, particularly in large time series datasets (such as those with tens of thousands of time steps). The authors do not provide a comprehensive analysis of the computational resources, which could affect the practical applicability of HDDI in real-world scenarios.

4. In the Historical Data Supplement Module, the authors employ a straightforward approach based on data differences to assess the similarity between two windows. Could this lead to erroneous expansion of the training data? In the ablation experiments, it is noted that randomly filled data consistently outperforms non-filled data. Does this suggest that even with incorrect filling, the model can still learn the correct distribution? Could the authors elaborate on the reasoning behind this phenomenon?

5. The number of baselines compared in the experiments is insufficient, and some of the advanced methods are missing. The authors could incorporate more recent imputation techniques, such as TimesNet[1] and ImputeFormer[2], to enhance the comparative analysis.

[1] Wu, Haixu, et al. "Timesnet: Temporal 2d-variation modeling for general time series analysis." In ICLR, 2023.

[2] Nie, Tong, et al. "ImputeFormer: Low rankness-induced transformers for generalizable spatiotemporal imputation." In KDD, 2024.

**Questions:**

As in weaknesses.

---

### Official Review · Reviewer_ahkL · 2024-11-09

**Soundness:** 2
**Presentation:** 2
**Contribution:** 1
**Rating:** 3
**Confidence:** 4

**Summary:**

This paper proposes HDDI method is innovative and shows promising results on the time-series imputation task.

**Strengths:**

1. The paper is well-structured and easy to follow;
2. The method is clearly illustrated by the figures;
3. The experiments conducted on five datasets are thorough, and the results are clearly presented;

**Weaknesses:**

I investigated the code and found the model backbone diff_models.diff_HDDI is exactly a CSDI model (diff_models.diff_CSDI in https://github.com/ermongroup/CSDI), but I did not see the authors mention this if I missed nothing. In my opinion, the authors propose a new historical data processing method to help train the model. I do not think this work makes enough contribution to the community.

**Questions:**

1. Did the hyperparameters of the baseline models get optimized? If so, please provide the details of the experiments regarding this point.
2. I didn't see baseline models in the code repository. I would like to learn about how the authors reproduced the baseline methods. Using their official code or other unified Python libraries (e.g. PyPOTS, Time-Series-Library)? It is important because data processing is quite different across imputation algorithms (their own implementations), while unified interfaces can ensure fairness in the experiments.
3. All experimental results are averaged over five trials. But why the standard deviation is missing? It is also a vital argument to evaluate the models;

Additional suggestions:
1. To improve the reproducibility of your results, please also list the configuration details of your development environment, e.g. Python package versions, which are missing in the provided code repository;
2. There is only one dataset processing script (i.e. Air Quality) in your code base. It would be better to share all of them to ensure the reproducibility;

---

### Note · Authors · 2024-11-26

I have read and agree with the venue's withdrawal policy on behalf of myself and my co-authors.